# Hybrid Prediction in Horticulture Crop Breeding: Progress and Challenges

**DOI:** 10.3390/plants13192790

**Published:** 2024-10-04

**Authors:** Ce Liu, Shengli Du, Aimin Wei, Zhihui Cheng, Huanwen Meng, Yike Han

**Affiliations:** 1Cucumber Research Institute, Tianjin Academy of Agricultural Sciences, Tianjin 300192, China; liuce2022@163.com (C.L.);; 2College of Horticulture, Northwest A&F University, Yangling 712100, China; 3State Key Laboratory of Vegetable Biobreeding, Tianjin 300192, China

**Keywords:** heterosis, hybrid prediction, marker-assisted selection, genomic prediction

## Abstract

In the context of rapidly increasing population and diversified market demands, the steady improvement of yield and quality in horticultural crops has become an urgent challenge that modern breeding efforts must tackle. Heterosis, a pivotal theoretical foundation for plant breeding, facilitates the creation of superior hybrids through crossbreeding and selection among a variety of parents. However, the vast number of potential hybrids presents a significant challenge for breeders in efficiently predicting and selecting the most promising candidates. The development and refinement of effective hybrid prediction methods have long been central to research in this field. This article systematically reviews the advancements in hybrid prediction for horticultural crops, including the roles of marker-assisted breeding and genomic prediction in phenotypic forecasting. It also underscores the limitations of some predictors, like genetic distance, which do not consistently offer reliable hybrid predictions. Looking ahead, it explores the integration of phenomics with genomic prediction technologies as a means to elevate prediction accuracy within actual breeding programs.

## 1. Introduction

Heterosis, the biological phenomenon in which hybrids produced through inter-specific or intra-specific hybridization surpass their parental lines in yield, growth vigor, viability, and stress resistance, is foundational to contemporary breeding systems, notably in horticultural crops [1,2]. To explain the phenomenon of heterosis, early breeders proposed three classical models [3]: (1) The dominance model, which suggests that harmful recessive mutations present in a homozygous form in the parents are masked and complemented in the hybrids, thereby manifesting heterosis. However, this model fails to account for the contribution of beneficial recessive alleles to heterosis [4]. (2) The overdominance model, which posits that heterosis arises from the heterozygous state of certain loci, rather than dominance relationships between alleles. Yet, this model cannot explain why modern inbred lines often outperform earlier generations of inbred lines [5]. (3) The epistasis model, which attributes heterosis to interactions between genes, including non-additive effects between loci [6]. With the rapid development of molecular biology, research based on single-gene loci, genome, transcriptome, metabolome, and microbiome has progressively uncovered the underlying mechanisms of heterosis from various perspectives [5]. However, despite these advances, no unified theory has yet been established. Nonetheless, heterosis has become a cornerstone of modern breeding theory and is widely applied in horticultural crop breeding practices [7].

In the practice of hybrid breeding, which involves generating a large number of hybrids and subsequently identifying the most promising candidates through extensive multi-year and multi-location field trials, logistical challenges are inevitable. For example, the diallel crossing method shows that an increase in the number of parent lines leads to an exponential growth in the number of potential hybrids, rendering comprehensive field evaluations at once impractical [8]. Accurate and timely prediction of hybrid performance could thus significantly streamline the selection process, enhancing breeding efficiency and expediting the breeding cycle. To this end, breeders have invested efforts in advancing hybrid prediction techniques in horticultural crops, making significant strides such as employing marker-assisted selection (MAS) for qualitative traits, while also navigating persistent challenges. This article delves into the latest developments in hybrid prediction, exploring both direct and indirect indices, and examines the ongoing challenges and avenues for future improvement, aiming to contribute insights into hybrid prediction research for horticultural crop breeding.

## 2. Hybrid Prediction Indices: Direct and Indirect

Hybrid performance evaluation indices are divided into two primary types, according to breeding goals: direct and indirect indices. Direct indices concern phenotypic information, including yield [9], viability [10,11], disease resistance [12], and stress resistance [13,14]. Direct indices are primarily used for comparisons among potential hybrids. For example, within a cohort of F_1_ hybrids, phenotypic data act as the basis for selecting promising hybrids. Depending on the genetic background, direct indices can categorize traits as either qualitative, controlled by single or a few genetic loci, or quantitative, influenced by numerous minor-effect loci. It is important to note that the line between qualitative and quantitative traits is not always clear [15].

Indirect indices, such as heterosis, involve comparing the F_1_ hybrid against its parents, employing metrics like mid-parent heterosis (MPH) and high-parent heterosis (HPH) (Figure 1), as well as comparisons among hybrids, for example, over-standard heterosis based on a benchmark line. Heterosis indicates the potential of hybrids to surpass their parental lines in yield or disease resistance, making it a vital evaluation index. Although heterosis is a widely acknowledged phenomenon in crop breeding, its underlying mechanisms are highly complex and not fully elucidated. Recent studies suggest that the manifestation of heterosis may be associated with mechanisms of single-parent expression, wherein hybrids activate a broader array of gene expressions compared to their parents, enhancing their adaptability to environmental conditions [16,17]. Further, metabolomic studies have revealed non-additive variations in proteins and metabolites related to photosynthetic and photorespiratory processes in hybrids relative to their parents [18,19], which could also serve as key predictors of heterosis. Interestingly, in plant–microbe interaction systems, the recovery of heterotic traits in hybrids appears to be critically influenced by beneficial microorganisms, including fungal symbiosis and bacterial colonization by auxin producers [20]. These complexity presents significant challenges in predicting heterosis, with diverse patterns emerging across various species and even within the same species for different traits [21,22].

## 3. Hybrid Prediction through Marker-Assisted Breeding

In horticulture crop breeding, qualitative traits often related to quality, such as the white or green fruit color of immature cucumbers (*Cucumis sativus* L.) and the purple or white curds of cauliflower (*Brassica oleracea* var. *botrytis*), are predominantly determined by a single or limited number of genetic loci [23,24,25,26]. Genetic defects, including male sterility mutations, serve as additional examples of qualitative traits [27,28,29]. Linkage and association analyses are pivotal in mapping these traits’ controlling genetic loci, paving the way for the development of co-dominant molecular markers, like simple sequence repeats (SSRs) and derived cleaved amplified polymorphic sequences (dCAPSs), which are closely linked to these loci. This process underpins MAS, allowing for the early prediction and selection of quality traits in potential hybrids (Figure 2). Moreover, MAS is instrumental in predicting and selecting quantitative traits like fruit yield [30] and disease resistance [31] governed by significant quantitative trait loci (QTL), including those associated with heterosis, such as the *SINGLE FLOWER TRUSS* (*SFT*) gene affecting fruit yield in tomatoes (*Solanum lycopersicum* L.) [9], and the *SUN* gene and its cucumber homolog *CsSUN*, which influence fruit shape [32,33]. MAS also addresses loci related to disease and abiotic stress resistance [34,35]. Despite MAS’s critical role in breeding, its capacity to capture and utilize minor QTLs in quantitative traits remains limited [36], highlighting a significant bottleneck in enhancing the predictive accuracy of MAS for these traits.

## 4. Hybrid Prediction through Genomic Prediction

Quantitative traits such as crop yield are significantly influenced by environmental factors and are typically controlled by a vast number of minor-effect genetic loci [37,38]. Traditional MAS faces challenges in accurately predicting quantitative traits, as the subtle effects of minor QTLs frequently fail to surpass statistical thresholds in QTL scanning, leading to their underutilization [39]. Nevertheless, these minor QTLs play a crucial role in influencing quantitative traits. Genomic prediction, introduced in 2001 and based on genome-wide high-density markers, addresses these challenges effectively [40]. It has since become integral to modern breeding strategies [41,42].

The process of genomic prediction initiates with the formation of a relatively small training population alongside a larger virtual test population. Subsequent steps involve gathering high-density genotypic and phenotypic data from the training population. Using these data, genomic prediction models are then trained to accurately estimate the impact of each genetic marker on the target trait or ascertain the breeding values of the potential hybrids. This methodology enables the precise prediction of the test population’s phenotype, relying on the genotype information as per the established model framework [42,43,44,45] (Figure 2). Typically, the molecular markers’ count (*p*) vastly outnumbers the hybrids (*n*) in the training population (*p* >> *n*), posing challenges in estimating marker effects or breeding values due to multicollinearity. To overcome this, strategies such as variable selection, coefficient shrinkage, and dimension reduction are implemented during model training to alleviate the issues of multicollinearity [46,47,48,49].

The core formula of genomic prediction models is defined as
(1)y=Xβ+Zu+ϵ
where y is the phenotype vector for the training population; β and u are vectors representing fixed and random effects, respectively, with u typically signifying marker effects or individual breeding values. The matrices X and Z are designated for fixed and random effects, respectively, where Z often stands for genotype or genetic relationship matrices; ϵ represents the residual vector of the model.

To address model collinearity and precisely estimate u, regularization methods are employed. These methods aim to minimize the sum of squared deviations while adding a penalty term under linear regression, including LASSO and ridge regression [50]. Bayesian techniques set an initial hypothesis for u, calculating the posterior probability of u using prior density and likelihood functions, such as BayesA, BayesB, and Bayesian LASSO [51]. Furthermore, a range of machine learning algorithms has been extensively incorporated into the genomic prediction workflow. Notable examples include random forests, rooted in decision tree theory [52]; support vector regression, based on support vector machine theory [53,54]; and deep learning models, utilizing neural network algorithms [55,56,57].

Despite the considerable potential of genomic prediction models for predicting quantitative traits, several factors can limit their predictive capability: (1) The genetic architecture of target traits: Models tend to perform better if the trait’s genetic architecture is consistent with the model’s assumptions. The control of traits by a small number of high-impact QTLs versus numerous minor-effect QTLs can significantly affect model accuracy [58]. (2) The size and representativeness of the training population: generally, larger and more representative training populations enhance model robustness [59]. (3) Trait heritability: higher heritability indicates a greater proportion of genetic variance relative to phenotypic variance, enabling genomic prediction models to capture more genetic variance for improved predictive capability [60]. (4) The measurement accuracy of phenotypic traits: controlling environmental factors and enhancing the accuracy of phenotypic measurements can reduce residuals and improve model predictive capability. (5) Marker density and linkage disequilibrium (LD): generally, model predictive capability improves with an increase in the number of SNPs, although gains are limited once marker density reaches saturation [61].

Heterosis is a phenomenon that is well acknowledged in plant breeding, leading to F_1_ offspring often surpassing the parental average in performance [62]. This deviation from the norm cannot be precisely captured by genomic prediction models that solely focus on additive effects. Incorporating predictors that account for non-additive effects, such as dominance and epistasis—referred to as non-additive models—has been shown to significantly enhance prediction accuracy [63]. For example, in a simulated pumpkin (*Cucurbita* spp.) breeding system, an additive-dominant model that included dominant effects improved predictive accuracy by 70% [64]. Similarly, in maize breeding, the incorporation of dominant effects into the genomic best linear unbiased prediction (GBLUP) model boosted the predictive accuracy by 16% to 26% [65], and in canola (*Brassica napus* L. subsp. *napus*) disease resistance breeding, an LMM that accounted for “additive × additive” epistasis increased the predictive power by up to 40% [66]. Additionally, the reproducing kernel Hilbert space (RKHS) model, adept at capturing non-additive effects, has seen wide application in genomic prediction breeding practices [67,68]. While non-additive models offer substantial improvements in predictive capability in certain contexts, they also risk overfitting due to their complex framework [63], highlighting the importance of careful optimization of non-additive effects.

Genome prediction was initially applied to animal breeding [69] and simulated datasets [61,70,71], and it began its gradual integration into plant breeding practices in the early 2010s [72]. Its application has since expanded across various crops including rice (*Oryza sativa* L.), maize (*Zea mays* L.), and wheat (*Triticum aestivum* L.), proving to be highly efficient and reliable for predicting quantitative traits [37,73,74]. In recent years, genome prediction has also found its way into horticultural crop breeding (Table 1). This approach has been implemented in various fruit crops, including apple (*Malus* × *domestica* Borkh.) [75,76], grapevine (*Vitis vinifera* L.) [77], strawberry (*Fragaria* × *ananassa*) [78,79], pear (*Pyrus bretschneideri* Rehd.) [80], cranberry (*Vaccinium macrocarpon*) [81], and blueberry (*Vaccinium corymbosun* L.) [82], as well as in several vegetable crops, such as tomato [83,84,85], cucumber [86], pepper (*Capsicum* spp.) [87], and cauliflower [88]. Additionally, genomic prediction has been utilized in the breeding of ornamental plants [89] and tea crops [90,91]. Although genomic prediction has not yet seen widespread use in horticultural crop breeding, its future application prospects appear very promising. Additionally, for polyploid crops like potatoes (*Solanum tuberosum* L.), algorithms based on allele dose effect theory have been developed, further advancing the breeding of polyploid crops [92,93]. As the cost of high-throughput sequencing continues to decline, the application of genomic prediction in horticultural crop breeding is set to become increasingly competitive.

## 5. Heterosis Prediction Based on Genetic Background Differences

Heterosis, which highlights the superior performance of F_1_ hybrids over their parents, acts as an indirect index for assessing hybrid vigor. Prior to the molecular biology era, crop breeding primarily depended on phenotypic selection, a technique characterized by its extensive history and limited efficiency throughout crop domestication [102]. The essence of heterosis stems from the genetic differences between parents. Initial heterosis prediction methods utilized physiological and biochemical indicators, such as isozymes, for identifying germplasm genetic diversity [103,104], and mitochondrial/chloroplast complementation tests to evaluate parental “affinity” through differences in homogenates [105]. Early genetic markers, including restriction fragment length polymorphism (RFLP), amplified fragment length polymorphism (AFLP), and random amplified polymorphic DNA (RAPD), facilitated the calculation of genetic distances among parents as a means to predict heterosis [106,107]. However, the predictive efficiency and accuracy of these initial approaches were limited due to sparse genetic background information and inconsistent reproducibility [108].

Advancements in sequencing technologies and the availability of crop genomes [109] have led to the development and increased application of genome-wide SSR markers in heterosis prediction studies (Table 2, [110]), attributed to their high density, co-dominance, and reliability. Although SSR-based genetic distance has demonstrated potential in maize breeding [111], its effectiveness appears to be crop-dependent [112,113]. The introduction of next-generation sequencing and SNP array technologies has significantly reduced genotyping costs, facilitating the detection of ultra-high-throughput SNPs. Explorations into SNP-based genetic distances and their correlation with heterosis have yielded variable results across different crops and traits. For instance, significant correlations were observed in wheat [114], moderate associations in pearl millet (*Pennisetum glaucum* L.) [115], and only weak correlations in upland cotton (*Gossypium hirsutum* L.) [116]. These outcomes suggest that while SNP markers provide dense information, their association with heterosis requires further investigation to ascertain their predictive value.

In addition, the theoretical framework of combining ability serves as a crucial index for elucidating quantitative genetics and heterosis performance, particularly within single-cross breeding systems. General combining ability (GCA) represents the average performance of a parent when crossed with various lines, highlighting the inbred parents’ performance. In contrast, specific combining ability (SCA) signifies the deviation of a specific hybrid from its parent’s GCA [131]. Combining ability can be derived from diallel or other mating designs [132,133]. Griffing (1956) described that the variance of GCA comprises additive and additive × additive genetic variances, while SCA’s variance includes non-additive genetic variances, such as dominance and epistasis. Recent studies have shown that SCA exhibits stronger correlations with heterosis than GCA [113,118,129], aligning with the hypothesis that heterosis is associated with non-additive genetic effects [134,135].

## 6. Why Is Genetic Distance Not Always Effective for Heterosis Prediction?

The utilization of genetic distance for predicting heterosis, though once popular [108], aligns with the observation that heterosis stems from genetic diversity between parents [3]. However, its predictive validity has been a subject of debate [136]. Despite the precision offered by high-density SSR/SNP markers in determining genetic distances, their correlation with heterosis prediction efficiency remains ambiguous [124]. Two primary concerns emerge:

(1) Limited Information from Genetic Distance: Classical algorithms like Nei’s, Edwards’, and Rogers’ distance [137] focus solely on marker number and allele frequency. Yet, heterosis is influenced by specific phenotypes governed by numerous QTLs, whose positions and effects vary across the genome, rendering genetic distances insufficient for comprehensive genetic insight (Figure 3, [118,138]. Furthermore, non-additive genetic effects, crucial for heterosis [139], are derived from the hybrids’ deviation from parental lines, which these distances fail to capture adequately.

(2) Complex Mechanisms Underlying Heterosis:The emergence of heterosis encompasses a myriad of intricate processes that extend beyond mere genetic contributions. These include epigenetic modifications such as DNA methylation, histone acetylation, and alterations in small RNA [140]. Moreover, changes in the transcriptome, proteome, and metabolome, which are not directly detectable through genome sequencing, also play significant roles [141]. Consequently, the reliance on genetic distance for heterosis prediction is diminishing in breeding practice, facing the need for more nuanced and multifaceted approaches to address these challenges.

This exploration underscores the necessity for advancements in heterosis prediction methodologies that consider the multifactorial nature of genetic and epigenetic influences, pointing towards a more integrated and comprehensive future in crop breeding research.

## 7. Challenges and Prospects for Heterosis Prediction in Horticulture Crops

The advancement of heterosis prediction is closely linked to research progress in understanding heterosis mechanisms. Despite significant achievements, several issues guide future research in horticulture crops:

(1) Sexual Reproduction’s Role: For crops like potatoes, predominantly propagated vegetatively, the gene pool’s enrichment through sexual reproduction is limited, posing challenges for heterosis utilization. An innovative strategy proposed by Sanwen Huang’s team for exploiting potato heterosis involves overcoming self-incompatibility in diploid potatoes and employing techniques like recombinant breeding and MAS selection to cultivate sexually reproducing diploid species with strong heterosis [142]. Additionally, severe domestication has led to reduced genetic diversity in some horticultural crops, necessitating the collection of more germplasm resources, including wild relatives, to enhance breeding gene pools’ genetic diversity [143]. Gene editing technologies like CRISPR/Cas9 offer precise target gene editing to generate superior alleles [144].

(2) Quantifying Difficult Phenotypes: In horticulture crop breeding, quality traits are crucial but often challenging to quantify, limiting phenotyping efficiency and accuracy. Trait information can be decomposed or redefined; for example, key chemical components can represent fruit aroma traits [145], and disease incidence can be more accurately reflected by converting traditional disease indexes into disease area ratios [146]. For visual phenotypes, like disease classification, replacing subjective human observation with high-throughput images or spectral information and developing deep learning algorithms can establish efficient phenotype collection systems [147,148].

(3) Complex Mechanisms of Heterosis: Despite systematic studies on the genome, transcriptome, metabolome, and epigenome, heterosis mechanisms remain partially understood [3]. Enhancing the explanatory power of current genomic prediction models requires considering more heterosis-related predictors in multi-omic predictions [73,149,150]. Additionally, environmental factors significantly affect phenotype/heterosis performance, necessitating strict control of environmental variables and conducting multi-year/location field experiments to accurately estimate genotype × environment interactions (G×E) [151,152].

(4) Beyond Heterosis in Breeding: Not all breeding strategies focus solely on heterosis; some aim for an “excellent-excellent combination” to achieve optimal phenotypes. Here, phenotypic selection and the combination of various traits through hybridization are crucial to compensate for parents shortcomings, breeding hybrids with comprehensive performance. Developing multi-trait genomic prediction systems is essential for improving breeding efficiency [153,154].

(5) Enhancing Genetic Diversity in Germplasm Resources: Previous studies have demonstrated that the degree of heterosis in hybrids is associated with the genetic diversity of their parental lines [155]. However, most modern cultivated horticultural crops, such as cucumber, have experienced severe domestication bottlenecks [25], which has significantly reduced their genetic diversity. To enhance the diversity of horticultural crop traits and improve resistance to diseases and environmental stresses in future breeding efforts, it is essential to collect a wider array of landraces, wild relatives, and closely related species. By revealing their genetic backgrounds and domestication histories, these resources can enrich the gene pool available for heterosis prediction, ultimately increasing the effectiveness of hybrid breeding strategies.

## Figures and Tables

**Figure 1 plants-13-02790-f001:**
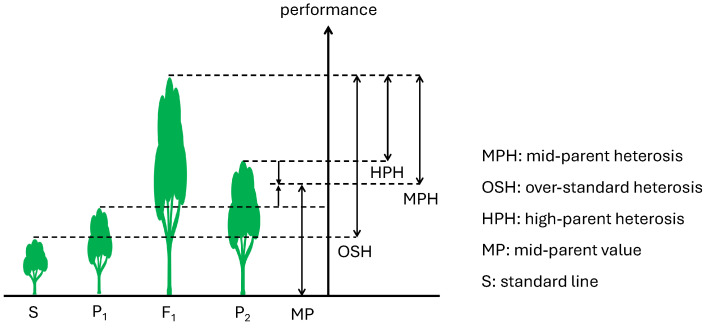
Common concepts of heterosis.

**Figure 2 plants-13-02790-f002:**
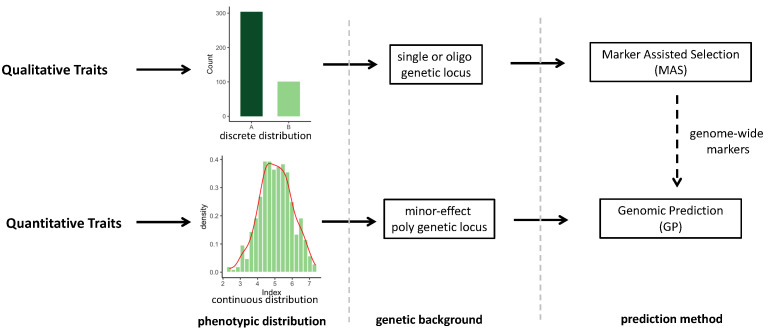
Genetic differences and prediction methods for qualitative and quantitative traits.

**Figure 3 plants-13-02790-f003:**
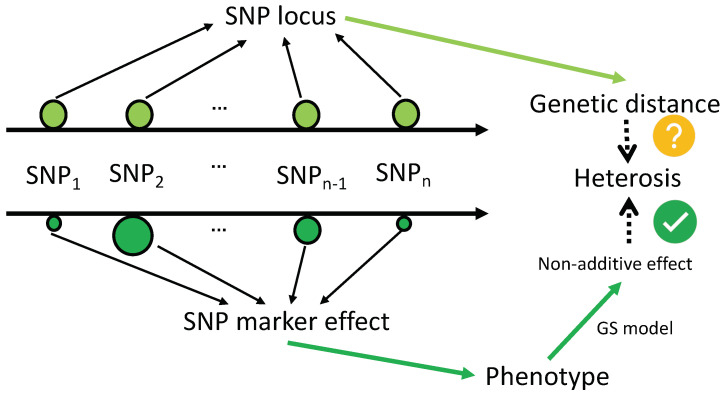
Predictors of heterosis: genetic distance and non-additive effects.

**Table 1 plants-13-02790-t001:** Published studies of genomic prediction in horticulture crop breeding.

Species	Traits	Population Size	Number of Markers	Models	Model Performance	Trait Heritability	Reference
apple	fruit size	977 individuals	7829 SNPs	BayesCπ	0.08–0.26 (PAc)	0.65 (h^2^)	[75]
pea	thousand seed weight	339 accessions	9824 SNPs	kPLSR, LASSO, GBLUP, BayesA, and BayesB	0.79–0.86 (PAc)	-	[94]
tomato	fruit weight	163 accessions	5995 SNPs	RR-BLUP	0.81 (PAb)	0.88	[83]
peach	Fruit weight	1147 plants	6076 SNPs	GBLUP	0.39–0.84 (PAb)	0.21–0.78 (h^2^)	[95]
strawberry	early marketable yield	1628 individuals	17,479 SNPs	GBLUP, BayesB, BayesC, BL, BRR, RKHS	0.42–0.63 (PAb)	0.29–0.43 (H^2^)	[78]
cauliflower	curd width	192 accessions	62,566 SNPs	RR-BLUP, GBLUP, BayesB	0.35–0.45 (PAb)	0.44 (H^2^)	[88]
rapeseed	plant height	203 inbred lines	24,338 SNPs	RR-BLUP	0.50 (PAc)	0.70 (H^2^)	[96]
potato	yield	571 clones	3895 SNPs	GBLUP	0.06–0.34 (PAc)	-	[92]
soybean	yield	483 lines	2647 SNPs	RR-BLUP	0.06–0.26 (PAb)	0.17	[97]
cassava	dry yield	290 clones	51,259 SNPs	GBLUP	0.42–0.50 (PAc)	0.62–0.78	[98]
pepper	fruit weight	351 accessions	18,663 SNPs	gblupRR, RR, LASSO, Elastic net, BL, EBL, BayesB, BayesC, RKHS, RF	0.79 (PAc)	0.97 (H^2^)	[87]
apple	fruit texture	537 genotypes	8294 SNPs	RR-BLUP	0.01–0.81 (PAc)	-	[76]
sugarcane	commercial cane sugar	3984 clones	26K SNP	GBLUP, GenomicSS, BayesR	0.36–0.57 (PAc)	0.87 (H^2^)	[99]
cowpea	100-seed weight	305 F_8:10_ RILs	32,059 SNPs	RR-BLUP	0.12–0.15 (PAc)	-	[100]
cucumber	commercial fruit yield	268 hybrids	16,662 SNPs	BRR	0.68–0.78 (PAb)	0.33–0.59 (H^2^)	[86]
strawberry	gray mold resistance	380 individuals	11,946 SNPs	GBLUP, RKHS, SVM	0.28–0.33 (PAc)	0.38 (h^2^)	[79]
tomato	yield	100 F_4_ generations	101,797 SNPs	RR-BLUP	0.73 (PAc)	-	[84]
rapeseed/canola	stem rot resistance	337 accessions	27,282 SNPs	RR-BLUP, BayesA, BayesB, BayesC, BL, BRR	0.60–0.61 (PAb)	0.69 (H^2^)	[101]
spinach	white rust resistance	346 accessions	13,235 SNPs	RR-BLUP, GBLUP, CBLUP, BayesA, BayesB, BL, BRR, RF, SVM	0.52–0.84 (PAc)	-	[31]
grapevine	mean berry weight	279 cultivars	32,894 SNPs	RR, LASSO	0.57 (PAb)	0.91 (H^2^)	[77]

PAc: prediction accuracy; PAb: prediction ability; H^2^: Broad-sense heritability; h^2^: narrow-sense heritability.

**Table 2 plants-13-02790-t002:** Published heterosis prediction studies based on genetic markers.

Species	Trait	Number of Markers	Population Size	*r* (GD: MPH)	*r* (GD: HPH)	*r* (GCA: MPH)	*r* (GCA: HPH)	*r* (SCA: MPH)	*r* (SCA: HPH)	Reference
Chinese cabbage	head weight	2,444,676 SNP	91 hybrids	0.17~0.21	−0.11~−0.09	-	-	-	-	[117]
cucumber	yield	16662 SNP	268 hybrids	0.11	0.01	0.38	0.43	0.65	0.61	[118]
melon	fruit weight	16 SSR	13 accessions	0.16	−0.20	-	-	-	-	[119]
pepper	fruit yield	6 AFLP	21 F_1_ hybrids	−0.14	−0.13	-	-	-	-	[120]
eggplant	yield	7335 SNPs	55 genotypes	0.11~0.19	-	-	-	-	-	[121]
pea	yield	14 SSR and 25 SRAP	45 F_1_ hybrids	0.26~0.33	0.11~0.41	-	-	-	-	[122]
carrot	total yield	12 RAPD and 9 AFLP	15 inbred lines and 34 hybrids	0.31~0.47	0.23~0.42	-	-	-	-	[123]
rapeseed	seed yield	7600 SNP	68 inbred lines and 132 hybrids	0.25	0.27	-	-	-	-	[124]
rapeseed	plant height	402 (SSR/SAP)	36 F_1_ hybrids	0.15	0.10	−0.43	−0.67	0.52	0.35	[113]
maize	grain yield	55 AFLP	136 F_1_ hybrids	0.41	0.28	-	-	0.47	0.31	[125]
maize	yield	1129 SNP	72 hybrids	-	0.37	-	-	0.48	0.31	[126]
wheat	grain weight	300 RAPD	76 F_2_ hybrids	0.10	-	-	-	−0.05	-	[127]
wheat	yield	4799 SNP	20 inbred lines and 100 hybrids	0.37	0.21	-	-	-	-	[114]
rice	grain yield	207 SSR	153 F_1_ hybrids	0.10~0.35	0.02~0.28	-	-	-	-	[128]
rice	grain yield	7098 SNP	33 hybrids	−0.06	−0.13	0.47	0.42	0.55	0.46	[129]
cotton	plant height	76,654 SNP	1128 hybrids	0.02	−0.05	-	-	-	-	[116]
cotton	plant height	198 SSR	1128 hybrids	0.01	−0.01	-	-	-	-	[116]
waxy corn	plant height	30 SSR	24 hybrids	−0.15	0.06	-	-	0.48	0.05	[130]
pearl millet	yield	56 SSR	147 lines	-	0.33	-	-	-	-	[115]
pearl millet	yield	75,007 SNP	117 lines	-	0.35	-	-	-	-	[115]

*r*: Pearson correlation coefficient. GD: genetic distance; MPH: mid-parent heterosis; HPH: high-parent heterosis; GCA: general combining ability; SCA: special combining ability.

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
