# Peer review of "Hybrid Prediction in Horticulture Crop Breeding: Progress and Challenges"

_plants, 2024, doi:10.3390/plants13192790_

Round 1

Reviewer 1 Report

Comments and Suggestions for Authors

The manuscript titled "Hybrid prediction in horticulture crop breeding: progress and challenges" takes a close look at the progress made in predicting hybrid outcomes for horticultural crops, with a special focus on how marker-assisted breeding and genomic prediction contribute to forecasting phenotypes. It also points out the limitations of certain predictors, like genetic distance, which may not always give reliable predictions for hybrids. This review is well-written. I think it would be suitable for publication in Plants. My only suggestion is to include more instances of hybrid prediction specifically related to horticultural crops.

Author Response

Comments 1: The manuscript titled "Hybrid prediction in horticulture crop breeding: progress and challenges" takes a close look at the progress made in predicting hybrid outcomes for horticultural crops, with a special focus on how marker-assisted breeding and genomic prediction contribute to forecasting phenotypes. It also points out the limitations of certain predictors, like genetic distance, which may not always give reliable predictions for hybrids. This review is well-written. I think it would be suitable for publication in Plants. My only suggestion is to include more instances of hybrid prediction specifically related to horticultural crops.

Response 1: Thank you very much for your recognition of our manuscript. In line with your suggestions, we have added several cases related to hybrid and heterosis prediction in horticultural crops. These additions can be found in lines 190 to 194 and in Table 2.

Reviewer 2 Report

Comments and Suggestions for Authors

The review presented by Ce Liu et al. focus on the possibilities of advancements in hybrid prediction for horticultural crops and highlights the development of prediction methods. The topic of this review is of a great importance, and all these comprehensive studies can help for breeders in planning and implementing breeding programs. This work is well-written and presented valuable and useful information.

I suggest minor revision of the ms. Authors can find my comments in the attached pdf file.

Author Response

The review presented by Ce Liu et al. focus on the possibilities of advancements in hybrid prediction for horticultural crops and highlights the development of prediction methods. The topic of this review is of a great importance, and all these comprehensive studies can help for breeders in planning and implementing breeding programs. This work is well-written and presented valuable and useful information.

I suggest minor revision of the ms. Authors can find my comments in the attached pdf file.

Response: Thank you very much for your acknowledgment of our review. We also believe that this article will provide meaningful references for breeding work in horticultural crops. Following your comments noted in the PDF, we have made the necessary revisions and responses. Below are the details of these updates.

Comments 1: line 53-54: Do you mean: Fruit appearance, appeal, taste texture, etc.?

In the text, there are some parts containing general statements. These statements are correct, but it would be great to insert examples (if there any), just like in line 144. (for example: Line 52. in „3. Hybrid Prediction through Marker-Assisted Breeding” chapter). Please, consider.

Response 1: We are conveying the point you mentioned. Certain appearance traits in horticultural crops, such as the white or green skin of cucumbers and the white or green curds of cauliflower, are quantitative traits controlled by single genes. We have clarified this information in the manuscript at lines 81 to 82.

Comments 2: Please, insert here some important agronomic traits!

Response 2: Thank you for your suggestion. We have added detailed information about specific traits related to yield and disease resistance in the manuscript, which can be found at lines 91 to 92.

Comments 3: line 65: Please, insert space!

Response 3: We apologize for the formatting issues in our manuscript. We have now added the necessary spaces. This correction has been made at line 95.

Comments 4: line 91: Please, use the template of Plants formatting the equation!

Response 4: Thank you for your valuable suggestion regarding the formatting of the equations. We have revised the equation format to align with the publication requirements of the journal Plants. This update has been made at line 121.

Comments 5: line 92: Delete 'Here', insert ....where, y is the....

Response 5: Thank you very much for your suggestions on improving the English phrasing. We have revised the relevant descriptions according to your advice, as reflected in line 122.

Comments 6: Table 2: The Table contains valuable information, but the paper is about horticultural crops. Maize, wheat, rice, rapeseed, cotton, pearl millet are not considered as horticultural crops but field crops.

Response 6: We appreciate your affirmation of the information in Table 2. As you pointed out, our manuscript primarily discusses the prediction of heterosis in horticultural crops. Following your suggestion, we have added some study cases on heterosis prediction in horticultural crops to Table 2. However, since most related research focuses on field crops, we have also retained part information on field crops to ensure the richness of the content. We hope that the revised table meets your approval.

Comments 7: - Does genetic diversity of horticultural crops affect heterosis prediction? Please, consider to insert the answer as point 5.

Response 7: Genetic diversity is crucial for hybrid breeding and predicting heterosis, so we fully agree with your comments on the impact of the genetic diversity of horticultural crops on heterosis prediction. Following your suggestions, we have added the section titled "Enhancing Genetic Diversity in Germplasm Resources," which now includes content from line 303 to line 312.

Reviewer 3 Report

Comments and Suggestions for Authors

The manuscript lacks a description of the main problem of heterosis breeding - our understanding of the mechanisms leading to such high performance in some hybrids. There is a reference to one article, discussing the problem (that is repeated throughout the text), but a more detailed description will make the manuscript more complete. My suggestion is that the introductory part should have at least some notion of the various theories that try to explain this phenomenon, preferably with a short explanation of their strengths and shortcomings. This will provide the background necessary for the arguments laid later in the text for the applicability of different approaches (i.e. Marker-Assisted Selection, Genomic Prediction, etc.). The expansion of this section can be easily compensated by greatly reducing other parts of the manuscript (see the comments below).

While having tackled the topic of our understanding of molecular basis for heterosis (Line 49), authors should include some more recent explorations of the idea (i.e. https://doi.org/10.1016/j.tplants.2024.07.018) as this field is rapidly evolving and a lot of new knowledge has been gathered since 2013.

Authors seem to confuse "lines" with "hybrids", which makes some of the statements difficult to follow. I.e. in Lines 36-37 they state that "within a cohort of F1 hybrids, phenotypic data act as the basis for selecting promising lines." As the term "line(s)" originates from the concept of "line of inheritance", it usually envisages one (or several) generation(s) from the same progeny that can be further propagated into consecutive generations with (almost) identical trait performance. Apparently, this is not applicable to the hybrids, therefore they remain a separate term with a very specific meaning - the F1 progeny only. As a consequence, the entire text must be revised to properly reflect that difference.

Part of the cation of Figure 2 is repeated, which needs to be corrected.

The text in Section 4 (Lines 70-137) is overly expansive on the discussion of genomic prediction based on additive models. This is unnecessary as the authors themselves acknowledge (Line 140-142) that these models have very limited (if at all existing) value for heterosis prediction. Therefore, this part can be significantly reduced - preferably to a couple of sentences explaining just that.

Comments on the Quality of English Language

English used is fine. Some terminological adjustments are needed.

Author Response

Comments 1: The manuscript lacks a description of the main problem of heterosis breeding - our understanding of the mechanisms leading to such high performance in some hybrids. There is a reference to one article, discussing the problem (that is repeated throughout the text), but a more detailed description will make the manuscript more complete. My suggestion is that the introductory part should have at least some notion of the various theories that try to explain this phenomenon, preferably with a short explanation of their strengths and shortcomings. This will provide the background necessary for the arguments laid later in the text for the applicability of different approaches (i.e. Marker-Assisted Selection, Genomic Prediction, etc.). The expansion of this section can be easily compensated by greatly reducing other parts of the manuscript (see the comments below).

Response 1: Thank you very much for your suggestion to enhance our manuscript's discussion on the mechanisms of heterosis. Understanding heterosis is indeed crucial for introducing the concept of heterosis prediction. In line with your advice, we have added a brief overview of the mechanisms behind heterosis formation in the introduction section (lines 18 to 29) and have further detailed recent research advances on heterosis mechanisms in lines 68 to 76.

Comments 2: While having tackled the topic of our understanding of molecular basis for heterosis (Line 49), authors should include some more recent explorations of the idea (i.e. https://doi.org/10.1016/j.tplants.2024.07.018) as this field is rapidly evolving and a lot of new knowledge has been gathered since 2013.

Response 2: We are especially grateful for the literature references you provided on the mechanisms of heterosis. Following your suggestions, we have incorporated these references into our manuscript, adding to the content on the progress in understanding heterosis mechanisms at lines 66-76. We hope that these revisions meet your approval and enhance the manuscript’s value.

Comments 3: Authors seem to confuse "lines" with "hybrids", which makes some of the statements difficult to follow. I.e. in Lines 36-37 they state that "within a cohort of F1 hybrids, phenotypic data act as the basis for selecting promising lines." As the term "line(s)" originates from the concept of "line of inheritance", it usually envisages one (or several) generation(s) from the same progeny that can be further propagated into consecutive generations with (almost) identical trait performance. Apparently, this is not applicable to the hybrids, therefore they remain a separate term with a very specific meaning - the F1 progeny only. As a consequence, the entire text must be revised to properly reflect that difference.

Response 3: Thank you very much for your valuable and professional revision suggestions. We acknowledge that there might have been some confusion between the concepts of "lines" and "hybrids" in our previous versions. Following your advice, we have reviewed and revised the relevant content throughout the manuscript, with special attention to Tables 1 and 2, to avoid any ambiguity. We apologize for any unprofessional expressions used previously.

Comments 4: Part of the cation of Figure 2 is repeated, which needs to be corrected.

Response 4: Thank you for your suggestions. We apologize for the issues in our formatting. The title information for Figure 2 has now been corrected at line 99.

Comments 5: The text in Section 4 (Lines 70-137) is overly expansive on the discussion of genomic prediction based on additive models. This is unnecessary as the authors themselves acknowledge (Line 140-142) that these models have very limited (if at all existing) value for heterosis prediction. Therefore, this part can be significantly reduced - preferably to a couple of sentences explaining just that.

Response 5: Thank you sincerely for your insightful suggestions regarding the structure of our manuscript. We acknowledge that the section on genomic prediction was initially quite detailed. In response to your feedback, we have streamlined this content by omitting specific details about the algorithms in the GS model and the sections on model cross-validation (lines 129-135, 138-140, and 145-153). This revision has helped tighten the narrative. Moreover, while you rightly pointed out the limitations of additive models for predicting heterosis, we maintain that the GS model remains vital for phenotype-based hybrid prediction. Therefore, we have opted to keep a refined framework of this content. We appreciate your guidance and hope that our revisions align with your expectations.

Comments 6: Comments on the Quality of English Language English used is fine. Some terminological adjustments are needed.

Response 6: Thank you for your professional suggestions on our English writing. We have systematically revised the manuscript to address the potential confusion between "lines" and "hybrids" that you previously pointed out. We hope that these revisions will meet with your approval and contribute to the clarity of the manuscript.